# Design of the VRLA Battery Real-Time Monitoring System Based on Wireless Communication

**DOI:** 10.3390/s20154350

**Published:** 2020-08-04

**Authors:** Rui Lu, Jiwu Lu, Ping Liu, Min He, Jiangwei Liu

**Affiliations:** 1Department of Electrical and Information Engineering, Hunan University, Changsha 410012, China; ruilu@hnu.edu.cn (R.L.); jiwu_lu@hnu.edu.cn (J.L.); pingliu@hnu.edu.cn (P.L.); hemin607@163.com (M.H.); 2Research Center for Functional Materials, National Institute for Materials Science (NIMS), 1-1 Namiki, Tsukuba, Ibaraki 305-0044, Japan

**Keywords:** condition monitoring, VRLA battery, wireless communication, state-of-charge

## Abstract

The VRLA (valve-regulated lead-acid) battery is an important part of a direct current (DC) power system. In order to resolve issues of large volume, complicated wiring, and single function for a battery monitoring system at present, we propose to build a novel intelligent-health-monitoring system. The system is based on the ZigBee wireless communication module for collecting voltage, temperature, internal resistance, and battery current in real-time. A general packet radio service (GPRS) network is employed for interacting data with the cloud-monitoring platform. The system can predict the remaining capacity of the battery combined with the software algorithm for realizing real-time monitoring of the battery’s health status and fault-warning, providing a basis for ensuring the safe and reliable operation of the battery. In addition, the system effectively integrates most of the circuits of the battery status collector onto one chip, which greatly reduces the size and the power consumption of the collector and also provides a possibility for embedding each VRLA battery with a chip that can monitor the health status during the whole life. The test results indicate that the system has the characteristics of real-time monitoring, high precision, small-volume, and comprehensive functions.

## 1. Introduction

The VRLA battery is an important part of the modern new energy field, and it has been the workhorse of uninterruptible power supply (UPS). With the development of the world economy, transportation, energy, communications, and other areas, the back-up VRLA battery systems are widely used [1]. As a series-connected component of the backup power supply, a single battery’s malfunction, due to various faults during operation, will result in the failure of the whole system. When the UPS should come to work, system failure may put not only the economy and social security at great risk but also human safety in an unconventional emergency. Thus, it is necessary to design an intelligent health-status monitoring system for the VRLA, to survey the battery parameters online, which is of great significance to ensure the safety and reliable operation of power supply equipment in various complex operation conditions.

According to the specific needs of different fields, there have been relevant battery monitoring systems applied to the photovoltaic standalone lighting system [2], renewable energy storage system [3], automobile power system [4], etc. The data transmission of existing battery monitoring systems is mainly through RS232, RS485, and other serial ports or the Ethernet [5,6]. However, the RS232 and RS485 serial ports have the disadvantages of short transmission distance, low transmission rate, and small data throughput. It is difficult to realize long-distance, fast, and large data communication. For a typical UPS module equipped in the rail transit system, at least tens of VRLA batteries are needed for a single UPS module. Therefore, the most repugnant thing for RS232 or RS485 health monitoring is the complicated wiring, which is basically prohibitive for its wide-applications. 

Nowadays, there are also some systems that receive and transmit data by using wireless communication. A battery monitoring system based on the Internet of things (IoT) is presented in Ref. [7] to monitor the operation and performance of batteries in a smart microgrid system. In Ref. [8], the authors demonstrated a complete system for monitoring an automotive battery. In the system, wireless sensors monitored each battery cell by measuring voltage and temperature. A central battery control unit was used to combine the cell measurements with current measurement and estimated the state of charge (SOC) and the state of health (SOH) of the battery. In Ref. [9], real-time monitoring of multiple lead-acid batteries based on the Internet of things is proposed and evaluated. The proposed system monitored and stored parameters that provide an indication of the lead-acid battery’s acid level, state of charge, voltage, current, and the remaining charge capacity in a real-time scenario. A wireless battery management system (BMS) based on Bluetooth technology is proposed in Ref. [10]. The implemented system performed similarly to a wired system in comparison tests while saving weight and significantly reducing failure points. A remote online monitoring system for the operation of the lead-acid battery group in telecommunication base stations is shown in Ref. [11]. Combining the general packet radio service (GPRS) communications and internet connections, the system realized data transmission between remote acquisition modules and a data services center. In Ref. [12], an electric vehicle battery management system based on a smart battery monitoring chip was designed, DS2438. It integrated the measurement of the battery’s temperature, voltage, current, and power as a whole, which not only simplified the circuit but also saved on system cost. The battery’s SOC can be easily estimated and displayed in this design. However, the existing systems are difficult to meet the requirements of most VRLA in real working environments in terms of volume, function, and so on.

At present, researchers have proposed a variety of capacity estimation methods for storage battery residual. However, it is not much to get popularization in real applications. The general estimation methods are mainly divided into traditional estimation methods, advanced intelligent estimation methods, and compound estimation methods [13]. The traditional estimation methods mainly include the open-circuit voltage method [14], discharge experiment method, resistance method [15], ampere-hour integral method [16], etc. Advanced intelligent estimation algorithms include the neural network [17], fuzzy control [18], support vector machine [19], Kalman filter [20], etc. The composite estimation methods are most commonly used because they can foster strengths and circumvent weaknesses, take the advantages of various algorithms, and maximize the accuracy of SOC estimation [21,22,23,24,25]. In the practical application, SOC is usually estimated by the ampere-hour integral method, which is simple, low cost, and easy to calculate. However, it is difficult to determine the initial SOC. Additionally, errors accumulate over time.

In this study, we propose to build a novel intelligent health monitoring system based on ZigBee wireless communication. It predicts the remaining capacity of the battery by historical data and a software algorithm, so as to realize the real-time monitoring and fault-warning of the battery health. In addition, the system integrates the microcontroller unit (MCU) circuit, wireless communication circuit, voltage, and internal resistance measurement circuits of the collector onto one chip reasonably and effectively, which would reduce the volume of the battery status collector greatly. We will demonstrate the hardware design of the modules, details of the proposed system along with software and algorithms, and the effectiveness of the developed hardware.

## 2. Design of the System

The intelligent health monitoring system based on ZigBee wireless communication is shown in Figure 1. It mainly consists of a main controller, collector, and cloud-monitoring platform. The collector transmits the voltage, temperature, internal resistance, and other parameters of the monitored battery to the main controller through the ZigBee communication mode [26,27]. The main controller stores the received data and communicates with the cloud monitoring platform through the GPRS network [11,28].

### 2.1. Design of the System Hardware 

The system hardware mainly includes the main controller and collectors. The main controller includes the data storage module, power management module, and communication (ZigBee, GPRS) module. The VRLA batteries are usually used in series-connection in a UPS system. The main controller also includes a total voltage detection module and a total current detection module. The total current detection module uses the mature and commonly used Hall sensor. The collector includes detection (voltage, temperature, and internal resistance) module, voltage regulator module, communication (ZigBee) module, etc. In the system, there is only one main controller, which corresponds to multiple collectors. The functions and communication between each component are schematically shown in Figure 2 and the structure diagram of the collector is shown in Figure 3. The major functions will be introduced in the following sub-sections.

#### 2.1.1. Voltage Sampling Module

The internal sigma-delta analog-to-digital converter (SDADC) of the collector MCU (STM32F373) is used to sample the terminal voltage of the VRLA battery. The SDADC module is a high-performance, low-power ADC with 16-bit resolution, and nine differential analog channels (with optional gain). During the converting of multiple channels, the conversion speed of each SDADC can be up to 16.6 ksps (thousands of samples per second). The conversion speed of each SDADC can reach 50 ksps if only one channel conversion is used. The sampling accuracy is not only related to the internal SDADC of MCU but also depends on our decent peripheral circuit design. The external conditioning and driving circuit of SDADC designed in this system are shown in Figure 4. With the help of them, signal filtering and instantaneous driving of analog input would be realized.

#### 2.1.2. Temperature Sampling Module

The temperature of the VRLA battery is constantly changing during the charging and discharging processes, which has a great influence on the residual capacity of the VRLA battery. Furthermore, high temperature is very dangerous for batteries, which can even trigger explosions. Therefore, it is vital to sample the temperature of the battery constantly. The system adopts a single-bus digital temperature sensor to realize real-time temperature monitoring. This sensor integrates temperature measurement and ADC into one, which makes temperature measurement simple and convenient to connect. Meanwhile, it occupies minimal space. The temperature measurement ranges from −55 °C to 125 °C with 12-bit default resolution. The sensor in our system chooses a two-wire connection mode instead of a three-wire connection mode. The parasitic power supply mode stores the energy in the internal capacitance when the signal line DQ is at a high level. It consumes the electric energy of the capacitance when the signal line is at a low level. The temperature measurement uses only one input/output port, which can reduce circuit wirings and power consumption. The parasitic power supply mode is shown in Figure 5.

#### 2.1.3. Internal Resistance Sampling Module

Internal resistance is an important parameter for monitoring battery capacity and health. The internal resistance measurements are DC measurement and alternating current (AC) measurement. The AC method can directly measure the internal resistance of the battery by injecting a small AC signal. However, it is easily affected by external noise, which results in a poor anti-interference ability and complex circuits. While the DC method is relatively simple to realize, a lot of heat will be generated in the measurement process, which may damage the circuit and even burn the board. If it is properly controlled, the DC method would have higher measurement accuracy. Here, the system adopts an improved DC method for the internal resistance measurement [29,30]. The sampling board MCU realizes the instantaneous discharge of R15 and R16 by PWM controlling the on and off of an N-metal-oxide-semiconductor (NMOS, Q3). The AD channel sampling is used to measure the voltage drops at both ends of discharge resistances. The internal resistance of the battery is measured by Ohm’s law. Temperature control protection is considered to solve the heat issue generated by large current during the discharge process. The design of the partial internal resistance sampling circuit is shown in Figure 6. In order to avoid frequent conduction of NMOS (Q3) in the discharge process, we replace the resistance R10 in its left driving circuit with a negative temperature coefficient (NTC) thermistor, which forms a voltage divider circuit with R17. When the temperature rises, the bipolar junction transistor (BJT, Q1) will gradually change from the saturation region to the cut-off region. The NMOS (Q3) will react immediately and finally cut-off.

#### 2.1.4. Chip Design of Acquisition Terminal

In order to meet the stringent volume requirements for the VRLA battery monitoring module in some application scenarios and explore the possibility of embedded monitoring chips in the future, this system integrates some collector functional circuits onto one single chip for reducing the collector volume vastly. The MCU control circuit, ZigBee module circuit, voltage sampling circuit, power regulator, and internal resistance circuit are integrated onto the same chip. After the circuit layout optimization, it has the characteristics of small size, easy loading and unloading, stable function, etc. However, since some parts of the system either need to be closer to the battery (such as temperature sensor and NTC thermistor) or needs to be outside-located (such as antenna and power supply), they are still not integrated at present. The acquisition terminal is shown in Figure 7.

### 2.2. Design of the System Software

After collecting voltage, temperature, internal resistance, and other data by the collector status monitoring system, they are transmitted to the main controller by the ZigBee meshing networking to connect the cloud monitoring platform by GPRS network for data processing and analysis.

#### 2.2.1. Design of Parameter Acquisition Software

During normal operation, the internal resistance changes slowly during the use of the battery, and the internal resistance measurement will discharge the battery. Although a very short period is enough, it’s not suitable to collect internal resistance frequently. If no acquisition instruction is issued data packets are collected and sent periodically. Meanwhile, the parameter thresholds are set according to the battery type. When the collected data exceed the thresholds, the battery fails and the system will alarm. The whole program flowchart is shown in Figure 8.

#### 2.2.2. Design of ZigBee Wireless Module Software

The network of ZigBee contains a coordinator node, routing node, and terminal node. There is only one coordinator node. It is embedded in the main controller. The ZigBee module in the collector contains both the routing node and the terminal node. During the initial network construction, the coordinator node generates a unique personal area network identity document (PAN ID) to build a network that does not conflict with other networks. The routing node and terminal node will search for the coordinator node and send a connection request. After the request passes, the routing node and the terminal node get their respective ID from the MCU of the collector and store them in the electrically erasable, programmable, and read-only memory (E2PROM). Once the communication is built, the network structure is fixed. The main controller will check the interval between the sampling time and the current time. If the set interval is exceeded, it will send the instructions for collecting data.

During the normal operation, the application layer of the coordinator node works in the passive triggering mode (except for automatic network maintenance): receiving relevant commands sent by the serial port of the main controller’s MCU, broadcasting them to the routing node and terminal node, and transmitting the feedback information to the main controller’s MCU to form a transparent mode. To minimize the power consumption, the activity of the routing node and terminal node is passive triggering in the whole process. To further reduce the power level, they are set to sleep mode according to the characteristics of the terminal node. The number of terminal nodes and routes is approximately equal.

Considering the security of the ZigBee network, the network adopts the centralized trust center for network authentication. The new network members are not able to join in the ZigBee networks while it is in default. Only devices with the same coordinator node as the global trust center link key (TCLK) can join the network. The TCLK will be exchanged when the device joins the network, and the coordinator node assigns the only one link key to the newly added device for encrypted communications. Keys and messages are encrypted in the application support sublayer (APS) layer during the transmission.

#### 2.2.3. Design of GPRS Data Sending and Receiving Software

The GPRS is the abbreviation of general packet radio service and has a wide range of coverage, which can meet the application scenarios of the VRLA battery under various working conditions, and truly realize “always online”. The selected GPRS module uses hypertext transfer protocol daemon (HTTPD) client mode to communicate with the web server. The module will pack and send the data to the GPRS wireless network, then transmit it to the monitoring platform on the cloud. The collected data will be observed on the user interface of the monitoring APP, and the data can be analyzed and processed. At the same time, the command can be sent to the GPRS module. The relevant data transmission and receiving software flow is shown in Figure 9. In addition, the GPRS network has its own mature network security mechanism, including the user authentication process, wireless access security mechanism, and GPRS Encryption Algorithm (GEA) in the process of data transmission.

#### 2.2.4. Remaining Battery Capacity Estimation

The state of charge (SOC) is an important index to measure the state of the remaining battery capacity. From the point of view of power, the U.S. advanced battery consortium (USABC) defined SOC in the book “electric vehicle battery test manual” as: under the certain discharge rate of the battery, the ratio between the remaining electricity quantity and the rated capacity under the same working conditions.
(1)$$SOC\left(t\right)=\left(1-\frac{Q}{C_{I}}\right)\times 100\%$$
where Q is the discharge power; $C_{I}$ is the released power of battery under the constant current. In case of variable current or the complex working conditions, the corresponding $C_{I}\;$ will change. Therefore, in the actual engineering applications, the battery rated capacity $Q_{N}$ is generally used to replace $C_{I}$.

By considering the practical application, the SOC is estimated by the ampere-hour integral method. This method is simple, low cost, and easy for measuring. While the SOC is affected by temperature, charging and discharging current, and other parameters. In order to estimate the original SOC value, we construct a three-dimensional fitting curve by measuring the SOC under different voltage and current under different charge-discharge ratios. As shown in Figure 10, we can obtain the corresponding relationship between voltage, current, and SOC during the discharge process of a 160 Ah VRLA battery from 0.1 C (16 A) to 1.5 C (240 A). Besides, the trend of the relationship between them at different discharge rates is similar and the voltage at the end of high current discharge is lower. The curves of other batteries with different capacities can be obtained in a similar way. We modify the influence factor parameters of temperature, discharge current, and battery health to obtain the SOC value by using the improved ampere-hour method formula. The traditional calculation formula is:(2)$$\begin{array}{c} SOC=SOC_{0}-\frac{1}{Q_{N}}\int_{0}^{t}Id\tau \end{array}$$
where the $Q_{N}$ is the rated capacity of the battery; *I* is the battery current; $SOC_{0}$ is the original value of SOC.

(1) Correction of charge–discharge rate influence

Peukert’s empirical formula describes the relationship between discharge current and discharge time of lead-acid battery under constant current. The specific expression is as follows: (3)$$\begin{array}{c} I^{n}t=K \end{array}$$

In the formula, I is the discharge current; t. is the discharge time; *n* related to the type of battery, VRLA battery generally takes about 1.3; K is a constant related to the active substance.

Under the condition of constant current, the discharge capacity formula of the battery at a certain time is Q=It, bringing it into the formula (3), one can get:(4)$$\begin{array}{c} Q=I^{1-n}K \end{array}$$

Discharging the battery with a standard discharge rate $I_{0}$ and arbitrary discharge rate $I_{r}$ respectively, after testing, the following equation can be obtained.
(5)$$\begin{array}{c} \{\begin{array}{c} I_{0}^{n}t_{0}=K\\ I_{r}^{n}t_{r}=K \end{array} \end{array}$$

In order to get *n*, take the logarithm for both sides of the above formula, then,
(6)$$\begin{array}{c} \{\begin{array}{c} nlgI_{0}+lgt_{0}=lgK\\ nlgI_{r}+lgt_{r}=lgK \end{array} \end{array}$$

So, it can be calculated that n is:(7)$$\begin{array}{c} n=\frac{lgt_{r}-lgt_{0}}{lgI_{0}-lgI_{r}} \end{array}$$

So, the equivalent current coefficient is obtained as follows:(8)$$\begin{array}{c} \eta_{i}=\frac{I_{0}^{1-n}}{I_{r}^{1-n}}={\left(\frac{I_{0}}{I_{r}}\right)}^{1-n} \end{array}$$

By bringing the equivalent current coefficient into Equation (2), the ampere-hour method with a preliminary amendment can be obtained.
(9)$$\begin{array}{c} SOC\left(t\right)=SOC_{0}-\frac{1}{Q_{N}}\int_{0}^{t}\eta_{i}Id\tau \end{array}$$

(2) Correction of ambient temperature influence

The influence of ambient temperature on the SOC of the battery is great. Whe the temperature is high, the diffusion rate of ions in the battery plate is fast. Meanwhile, the conductivity of the electrolyte will increase. Enhancement of the temperature will increase the concentration of the battery electrolyte. Therefore, it is necessary to correct the ambient temperature. The following empirical formula is used to describe the relationship between temperature and capacity.
(10)$$\begin{array}{c} C=C_{25}\left[1+\alpha_{T}\left(T-25\right)\right] \end{array}$$

The $\alpha_{T}$ is the temperature coefficient, which is constant in the range of 0.003–0.01; T is the electrolyte temperature; C is the capacity; $C_{25}$ is the capacity at 25 °C; the temperature compensation coefficient of the battery is shown in the following equation:(11)$$\begin{array}{c} \eta_{T}=\frac{C_{25}}{C}={\left[1+\alpha_{T}\left(T-25\right)\right]}^{-1} \end{array}$$

By bringing the temperature compensation coefficient into Equation (9), the ampere-hour method can be further modified.
(12)$$\begin{array}{c} SOC\left(t\right)=SOC_{0}-\frac{1}{Q_{N}}\int_{0}^{t}\eta_{i}\eta_{T}Id\tau \end{array}$$

(3) Correction of battery health influence

Battery health is also known as battery aging. It means that with the increase in the number of battery cycles, the performance of the battery will decline and the service life will become shorter. As the number of cycles increases, the output capacity of the battery will decrease. In this study, the number of cycles is used to reflect battery life indirectly. The main factors that could affect battery life are: the falling off of the active material of the battery plate or the reduction of the surface area of the active material; the internal short circuit of the battery; the damage of the diaphragm, etc. Therefore, the battery capacity (*N*) corresponding to the number of charge and discharge is obtained through the multiple charge and discharge experiments of the battery in this study. The cycle number compensation coefficient ($\eta_{N})$ can be obtained by comparing the capacity with the rated capacity. Some of the data are shown in Table 1.

By bringing the cycle number compensation coefficient $\eta_{N}$ into Equation (12), the ampere-hour method can be further modified.
(13)$$\begin{array}{c} SOC\left(t\right)=SOC_{0}-\frac{1}{Q_{N}\eta_{N}}\int_{0}^{t}\eta_{i}\eta_{T}Id\tau \end{array}$$

## 3. Results and Discussion

For a whole system functions test, we selected two types of VRLA batteries (2 V 160 Ah, 2 V 300 Ah) for different charge and discharge ratios (0.1 C, 0.2 C, ..., 1 C). In the initial state, the sampling periods of voltage, current, temperature, and internal resistance are 10, 10, 10, and 30 s, respectively. The experimental batteries and test sites are shown in Figure 11 and Figure 12, respectively.

When six VRLA batteries (2 V, 300 Ah) are discharged at a constant current of 48 A to the termination voltage of 1.8 V, the relevant measurement data of one battery are shown in Figure 13. At the beginning of the constant current discharge, the terminal voltage amplitude of the battery shows a linear sharp drop, which is caused by the linear change of ohmic internal resistance. In addition, due to the thermal effect of the current, the temperature of the battery will rise slightly. The dynamic balance will be maintained after 80 min due to the stable heat dissipation of the battery itself. With the progress of discharge, the change rate of terminal voltage decreases, which is caused by the polarization effect inside the battery. Resistance and capacitance can be used to describe the impedance of ions during the transport of electrode reaction. Due to the system measurement error, the terminal voltage will fluctuate during the discharge process. In the later stage of discharge, because of the increasing resistance of chemical reaction and electrolyte concentration change, the internal resistance rises rapidly after around 220 min. Since the temperature also rises, and the terminal voltage drops rapidly to the termination voltage with the discharge [31,32]. In addition, we can also observe the battery parameters in real-time on the monitoring platform software as shown in Figure 14. If the parameter threshold is exceeded, an alarm will be released.

We have also calculated the measurement data of 50 VRLA batteries (2 V, 160 Ah) at a different discharge rate. The absolute error values of each parameter were continuously measured and counted, and the relative error was calculated, as shown in Figure 15. In measurement, the maximum average relative error of voltage is 0.31%, the temperature is 1.82% (±0.5 °C), and the internal resistance is 1.76%. It can be observed that the error fluctuation of temperature and voltage is relatively small, and the fluctuation of internal resistance is slightly larger, which is caused by the dynamic change of internal chemical reaction when the VRLA battery is working. The test errors meet the requirements of practical engineering application and verify the reliability of relevant design schemes. 

The system can also measure SOC accurately according to the improved SOC estimation method. We tested 50 VRLA batteries (2 V, 160 Ah) with unknown initial capacity for SOC estimation. The maximum absolute error of the test sample is less than 5%, and the test values fluctuate slightly near the accurate values. The test results meet the needs of engineering applications. The comparison diagram of test results is shown in Figure 16, in which the accurate value of the battery capacity is obtained through a discharge experiment.

We have compared our proposed system with existing systems such as the BMM from Gold Electronic and the iBattery Solution from Huawei, as shown in Table 2. Compared with the other two systems, the system proposed in this study has high measurement accuracy and simple wiring. Since the GPRS network with wide coverage is used, the proposed system is always online. Additionally, because most of the circuits of the battery collector are integrated onto one chip, it has a small volume. Therefore, the proposed system is suitable for more complex application scenarios.

## 4. Conclusions

Here, we present a VRLA battery online monitoring system based on ZigBee and GPRS technology. The collector collects the voltage, temperature, internal resistance, current, and other parameters of the battery, then transmits them to the main controller by the ZigBee group network, and then transmits them to the cloud monitoring platform through a GPRS network. Meanwhile, it can set the parameter thresholds according to the actual demand, timely alarm the abnormalities of the battery, and improve the original SOC calculation parameters to improve the precision of the prediction. The experimental results demonstrate that the system has a higher measurement accuracy by comparing the measurement results with the relevant measuring instruments. Moreover, by integrating most of the collector’s parts onto the same chip, the volume of the whole system is only 50% compared with the original one. It provides the possibility of embedding a chip that can monitor the whole life cycle health status before each VRLA battery is put into use in the future.

## Figures and Tables

**Figure 1 sensors-20-04350-f001:**
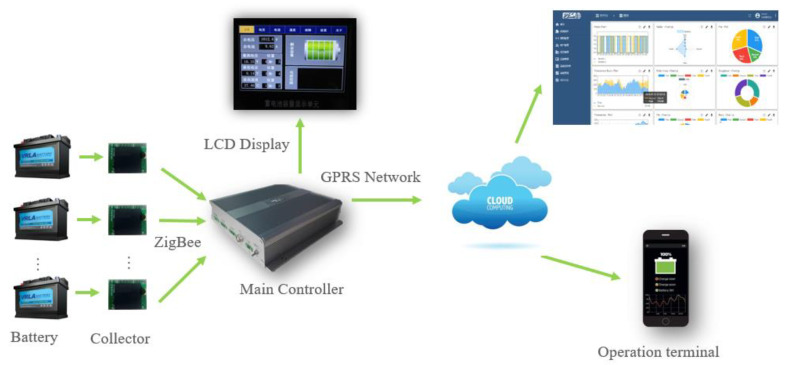
Architecture of the VRLA battery intelligent monitoring system.

**Figure 2 sensors-20-04350-f002:**
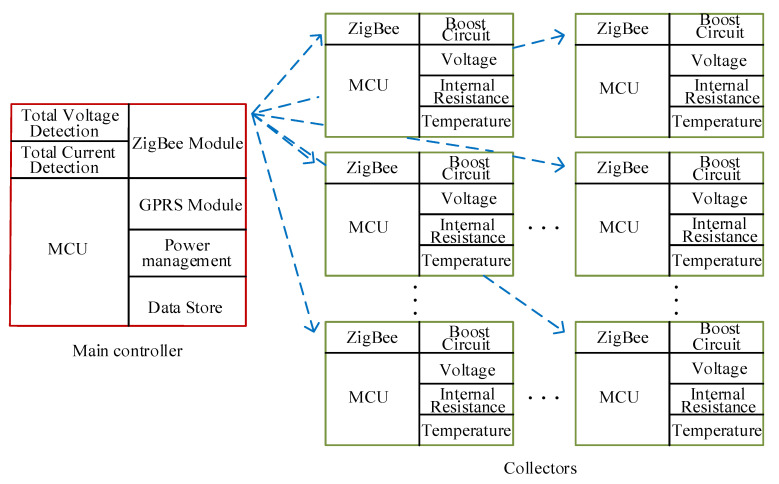
Functions and communication between the main controller and collectors.

**Figure 3 sensors-20-04350-f003:**
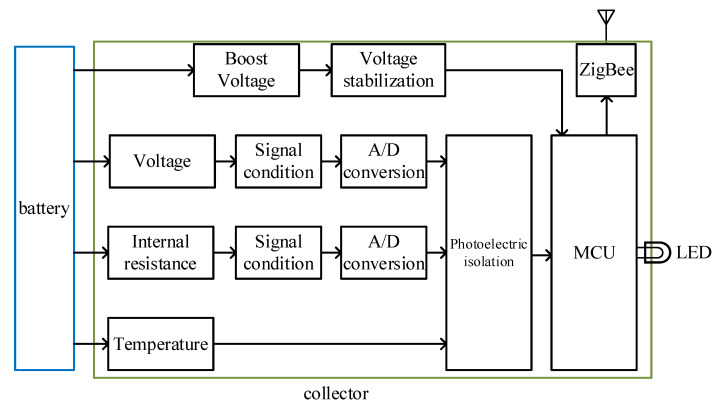
Structure diagram of the collector.

**Figure 4 sensors-20-04350-f004:**
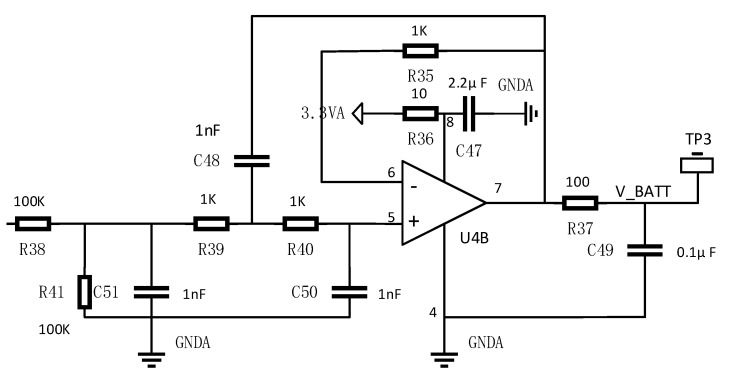
External conditioning and driving circuit of SDADC.

**Figure 5 sensors-20-04350-f005:**
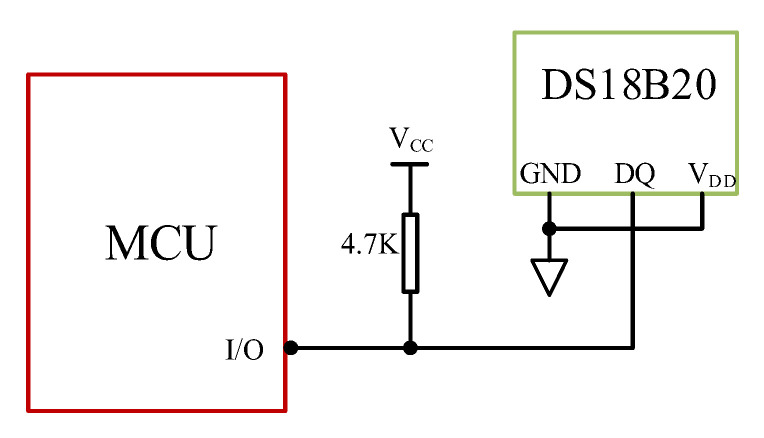
Diagram of DS18B20 parasitic power supply.

**Figure 6 sensors-20-04350-f006:**
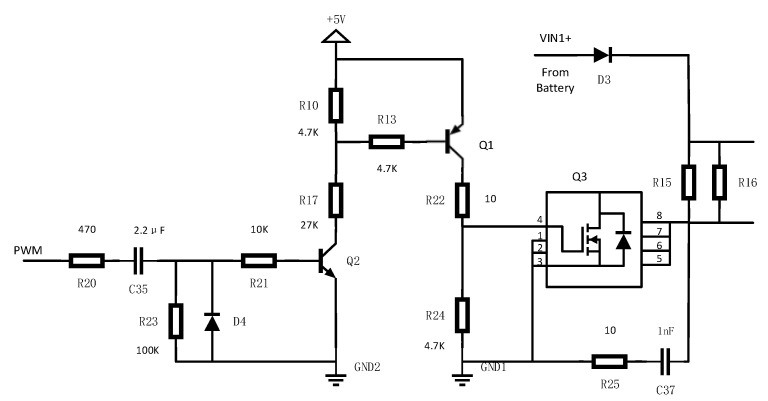
Partial internal resistance sampling circuit.

**Figure 7 sensors-20-04350-f007:**
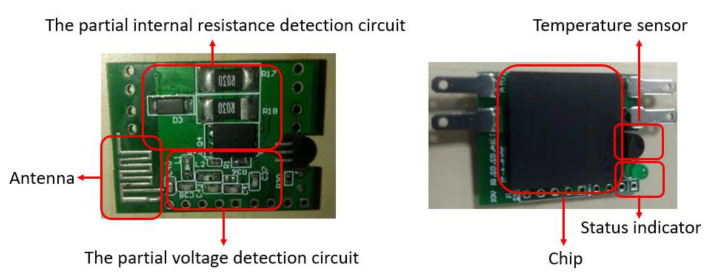
The design diagram of the collector.

**Figure 8 sensors-20-04350-f008:**
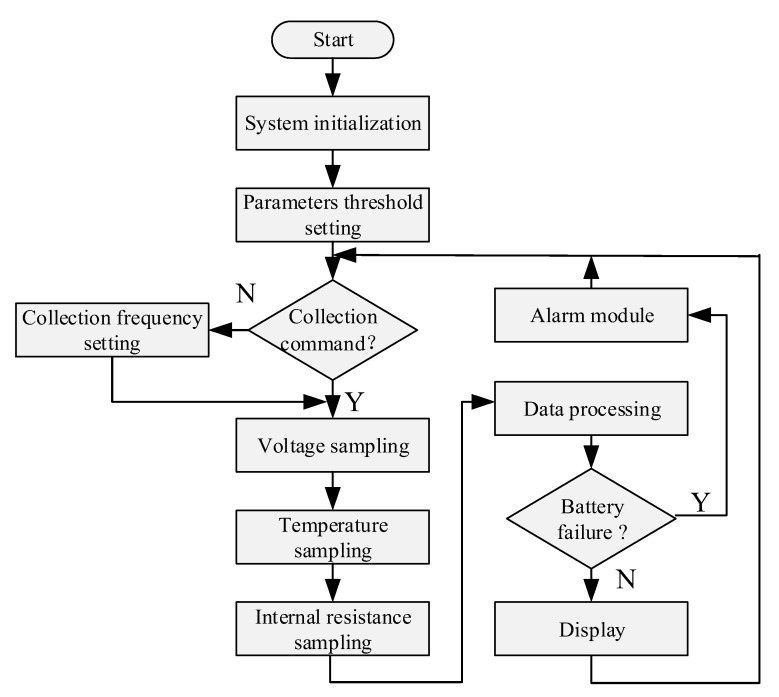
Flow chart of parameter acquisition software.

**Figure 9 sensors-20-04350-f009:**
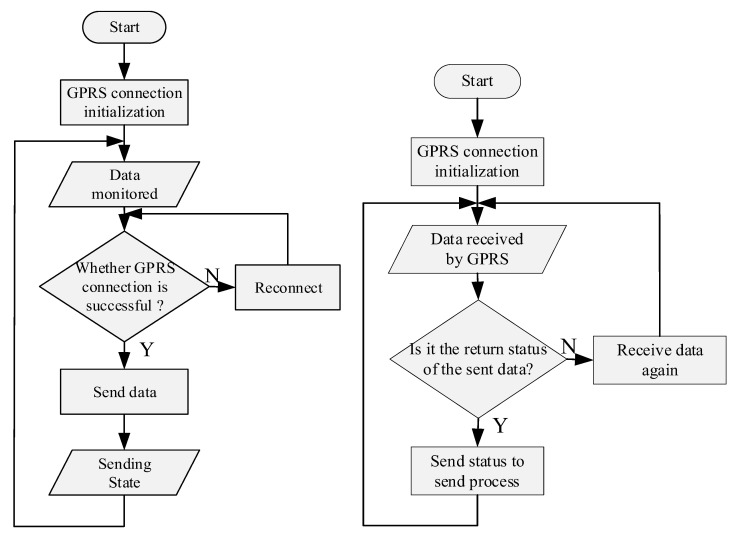
The flow chart of GPRS data sending and receiving software.

**Figure 10 sensors-20-04350-f010:**
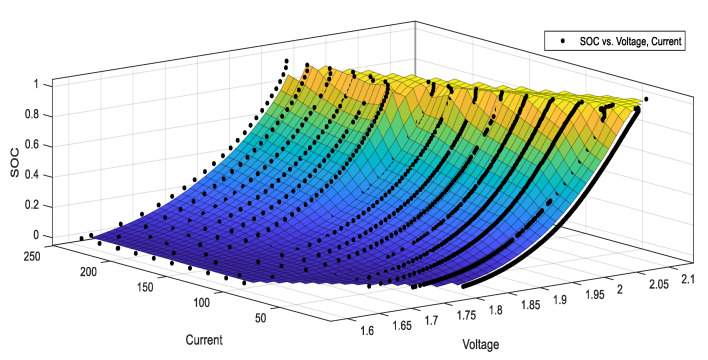
Fitting curve of voltage, current, and SOC.

**Figure 11 sensors-20-04350-f011:**
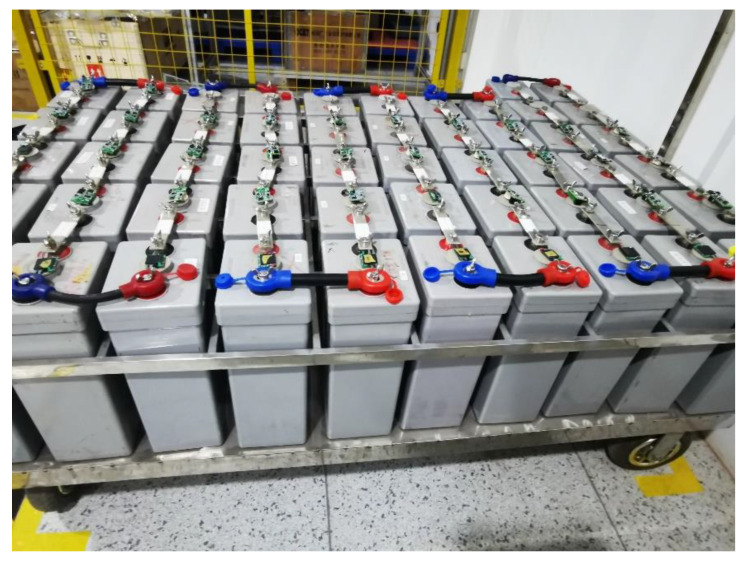
Physical picture of experimental batteries.

**Figure 12 sensors-20-04350-f012:**
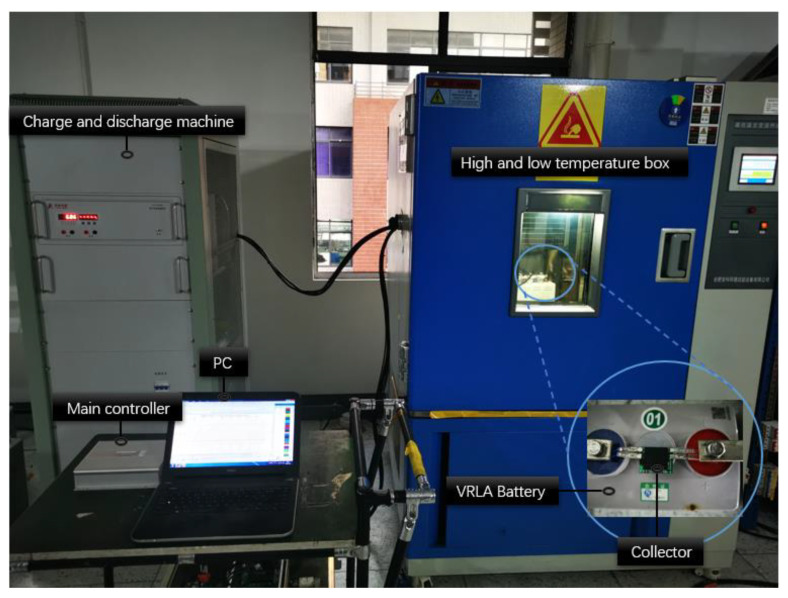
Picture of experiment and test site.

**Figure 13 sensors-20-04350-f013:**
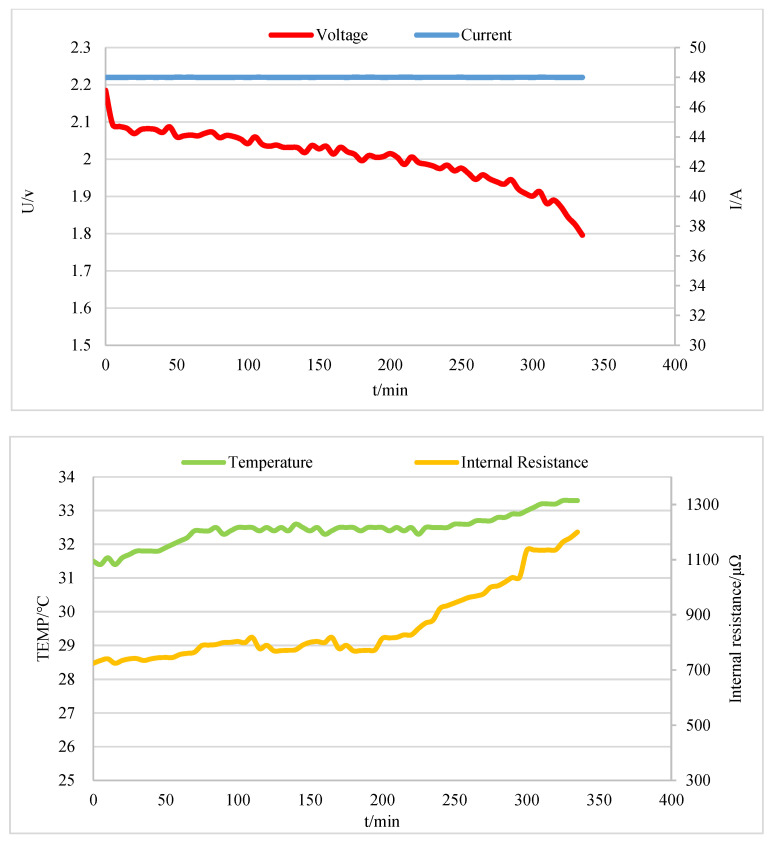
Voltage, current, temperature, and internal resistance measured by the system.

**Figure 14 sensors-20-04350-f014:**
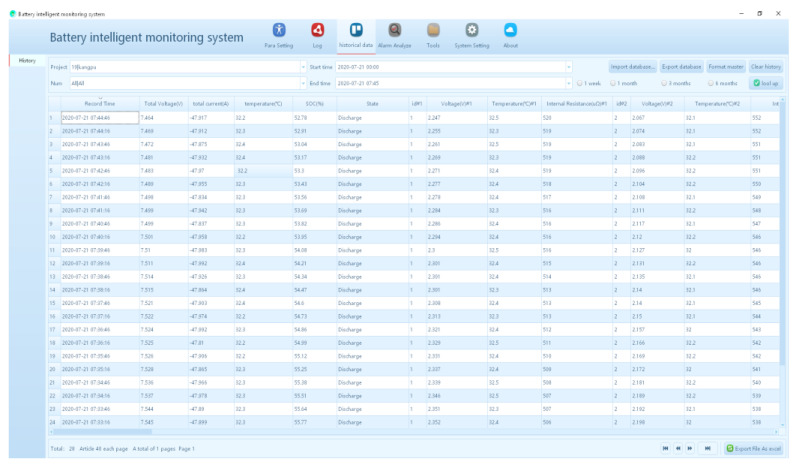
Software interface of the VRLA battery intelligent monitoring system.

**Figure 15 sensors-20-04350-f015:**
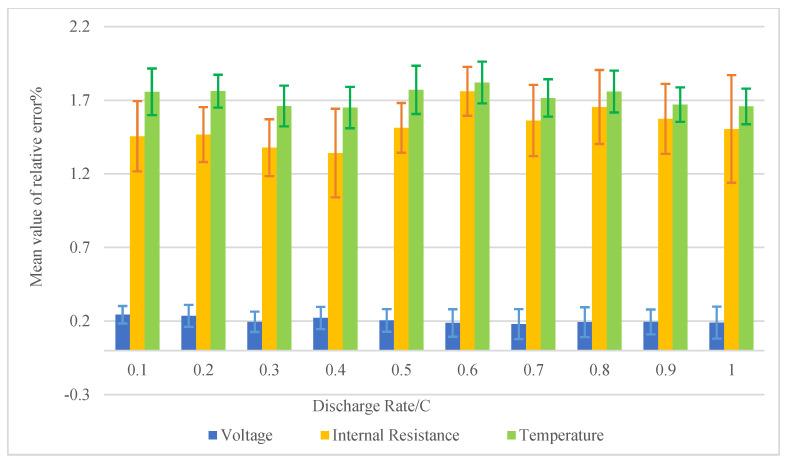
The statistics of the average relative error of each parameter measurement in the experiment.

**Figure 16 sensors-20-04350-f016:**
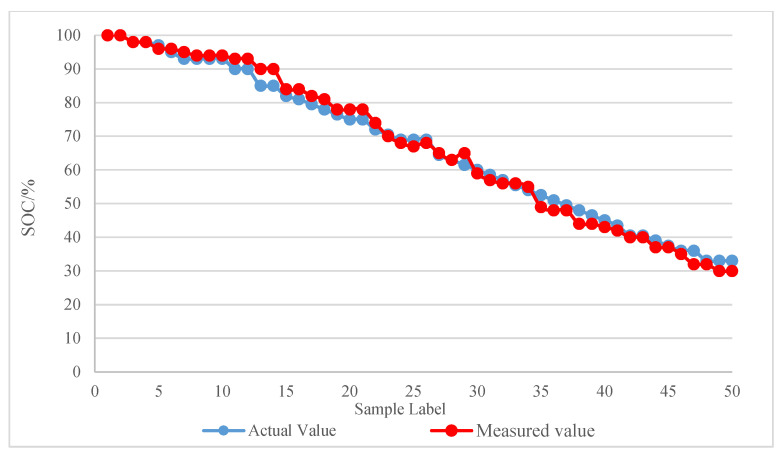
Comparison of SOC test results.

**Table 1 sensors-20-04350-t001:** Cycle number compensation coefficient $\eta_{N}$.

Number of Cycles (N)	$\eta_{N}$	Number of Cycles (N)	$\eta_{N}$
0	1.00	180	1.02
20	1.06	200	1.01
40	1.08	220	1.00
60	1.07	240	0.99
80	1.06	260	0.98
100	1.05	280	0.97
120	1.05	300	0.95
140	1.04	400	0.87
160	1.03	500	0.83

**Table 2 sensors-20-04350-t002:** Comparison of proposed system with existing devices.

	BMM Gold Electronic [33]	iBattery Huawei [34]	Proposed System
**Parameters**	Voltage (<0.2%)	Voltage (<0.5%)	Voltage (<0.35%)
	Current (<1%)	Current (<1%)	Current (<1%)
	Temperature (±0.5 °C)	Temperature (±0.5 °C)	Temperature (±0.5 °C)
	Internal Resistance (<2.5%)	Internal Resistance (<2%)	Internal Resistance (<2%)
	SOC (<5%)	SOC (<10%)	SOC (<5%)
**Communication Mode**	RS485	Zigbee	Zigbee/GPRS
**Wire**	Complex	Simple	Simple
**Real-Time**	Yes	Yes	Always Yes
**Remote Monitor**	No	No	Yes
**Integration**	Low	Low	High

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
