# Peer review of "Design of the VRLA Battery Real-Time Monitoring System Based on Wireless Communication"

_sensors, 2020, doi:10.3390/s20154350_

Round 1
Reviewer 1 Report
This paper proposed a battery monitoring system. This system is useful for increasing situational awareness of batteries and improving energy storage management. Comments are as follows.
- It is suggested to summarize the key innovations of the developed monitoring system, as well as the reasons why it has better performance compared with some existing products.
- Is there any cyber security concerns or considerations on this system? It is suggested to discuss it for the system.
Reviewer 2 Report
- The paper presents the design of a VRLA Battery Real-time Monitoring System based on ZigBee wireless communication module. The experimental results demonstrate that the system has a high measurement accuracy.
- The paper clearly identifies the scope and the contribution. It highlights the results and experiments. The authors give enough information for further implementation of their proposed Real-time Monitoring System.
- If you care to explain figure 10.
- In the abstract you specify that you provide a software algorithm for realizing real-time monitoring of the battery's health status and fault-warning, but you don’t give details about it later on in the paper.
- There are minor misspelling/rephrasing errors like:
39-40 rephrase
74 it can/they can
88 would reduces
125 which affects the residual capacity of VRLA battery greatly.
133-132 The sensor in our system chooses two-wire system mode to instead of three-wire mode.
195-196 Once the communication is built, the network structure is fixed with different collectors have unique PAN ID.
282 Battery health, also known as battery aging.
